# Effect of Na_2_CO_3_ on the Microstructure and Macroscopic Properties and Mechanism Analysis of PVA/CMC Composite Film

**DOI:** 10.3390/polym12020453

**Published:** 2020-02-14

**Authors:** Jufang Zhu, Qiuying Li, Yanchao Che, Xingchen Liu, Chengcheng Dong, Xinyu Chen, Chao Wang

**Affiliations:** 1School of Material Science and Engineering, East China University of Science and Technology, 130 Meilong Rd, Shanghai 200000, China; zhujufang1995@163.com (J.Z.); 18721503568@163.com (X.L.); 18621039859@163.com (C.D.); nagewo1314@163.com (X.C.); chaowang826@163.com (C.W.); 2Shanxi Novofluo New Material Science and Technology Co., LTD, Jinzhong 030600, China; liqy@ecust.edu.com

**Keywords:** PVA, CMC, Na_2_CO_3_, film

## Abstract

Polyvinyl alcohol (PVA)/carboxyl methyl cellulose sodium (CMC)/Na_2_CO_3_ composite films with different contents of Na_2_CO_3_ were prepared by blending and solution-casting. The effect of Na_2_CO_3_ on the microstructure of PVA/CMC composite film was analyzed by Fourier-transform infrared (FTIR) spectroscopy, X-ray diffraction (XRD), differential scanning calorimetry (DSC), and atomic force microscopy (AFM). Its macroscopic properties were analyzed by water sorption, solubility, and dielectric constant tests. The results show that the microstructure of PVA/CMC/Na_2_CO_3_ composite films was different from that of PVA and PVA/CMC composite films. In addition, compared to PVA and PVA/CMC composite films, the water sorption of PVA/CMC/Na_2_CO_3_ composite films relatively increased, the solubility in water significantly decreased, and the dielectric properties significantly improved. All these results indicate that the hydrogen bonding interaction between PVA and CMC increased and the crystallinity of PVA decreased after the addition of Na_2_CO_3_. This was also a direct factor leading to increased water sorption, decreased solubility, and enhanced dielectric properties. The reaction mechanism of PVA, CMC, and Na_2_CO_3_ is proposed to further evaluate the effect of Na_2_CO_3_ on the microstructure and macroscopic properties of PVA/CMC/Na_2_CO_3_ composite films.

## 1. Introduction

With increasing difficulties in terms of waste disposal and global warming, severe environmental problems are raising concerns all over the world. In addition, petroleum shortage is also a serious threat throughout the world. Considering these two problems, all countries are trying to develop environmentally friendly materials from nonpetroleum resources. Biopolymers are considered as an ideal alternative to traditional plastic materials owing to their advantages such as biodegradability, nontoxicity, and potential widespread applications; at present, a lot of research is focused on biodegradable materials. Among the biopolymers, cellulose and its water-soluble derivatives are mostly preferred because of the abundance of cellulose on earth and relatively low cost of cellulose derivatives compared to synthetic polymers [1].

Carboxyl methyl cellulose sodium (CMC) is an anionic, linear, and water-soluble polymer; CMC is generally formed by the carboxymethylation of hydroxyl groups in cellulose [2]. CMC has strong water solubility, hydrophilicity, and chemical reactivity because of the presence of hydroxyl and polar carboxylate groups. Moreover, it also has many other characteristic properties such as biocompatibility, renewability, nontoxicity, and biodegradability [3,4,5]. In addition, CMC has many excellent features such as film formation, emulsification, suspension, water retention, and bonding properties. Therefore, it is the most preferred polymer for food, medicine, detergent, paint, paper, flocculant, sizing agent, textile, and other industries [6,7,8,9,10,11]. On the other hand, polyvinyl alcohol (PVA) is a synthetic water-soluble polymer with many applications including textile pulps, adhesives, coatings, dispersants, and a series of medical materials with medical functions owing to its excellent adhesion, flexibility, and film-forming properties [12,13]. PVA is very suitable for blending with natural polymers such as CMC because of its biocompatibility and water solubility [14]. Owing to the presence of hydroxyl groups in both PVA and CMC, stronger intermolecular hydrogen bonds can be formed. Films made from blends of biopolymers generally exhibit more excellent properties compared to films made from biopolymers alone. Therefore, studies on PVA and CMC blend films were a hot topic in recent years. For example, El-Sayed et al. conducted differential scanning calorimetry (DSC), thermogravimetric analysis (TGA), and dielectric spectroscopy of blends of PVA and CMC with different compositions [15]. Zhang et al. controlled the sorption and permeability of PVA/CMC composite films through cross-linking, a potential coating for controlled release of fertilizers [16].

Extensive studies showed that the water sorption and solubility of PVA/CMC composite films have great significance for their applications. At present, some studies showed that different pH environments have a certain effect on the water sorption and solubility of composite films. Ibrahima et al. immersed PVA/CMC and PVA/cellulose composite films in solutions of different pH to evaluate the effect of pH on the solubility of films; the results showed that the solubility of composite films was the lowest at pH 10 [17]. Bajpai et al. studied the swelling ratio of CMC/starch composite films in different pH solutions; the swelling ratio constantly increased with increasing pH [18]. However, these studies only evaluated the water sorption of composite films in different pH solutions, whereas they did not systematically study the microstructure and other macroscopic properties of composite films in different pH solutions.

In this study, PVA and CMC were used as a biopolymer matrix, and Na_2_CO_3_ was used as a filler to evaluate its effect on the microscopic and macroscopic properties of PVA/CMC composite films. A reaction mechanism of Na_2_CO_3_ with PVA and CMC is proposed, providing a theoretical basis for improving the water sorption, solubility, and dielectric properties of PVA/CMC composite films and promoting their application in biodegradable superabsorbent resins and polymer electrolytes.

## 2. Materials and Methods

### 2.1. Materials

PVA (>99%, 87–89% hydrolyzed) and Na_2_CO_3_ (99.5%) were purchased from Shanghai Aladdin Biochemical Technology Co., Ltd., China. CMC (≥99.5%, Degree of substitution: 0.7) was purchased from Shanghai Yuanye Biotechnology Co., Ltd., China. All chemicals were used as received without any further purification. Deionized water was used, with a conductivity of 0.0556 μs/cm at 25 °C. The structural formula of PVA is as follows:
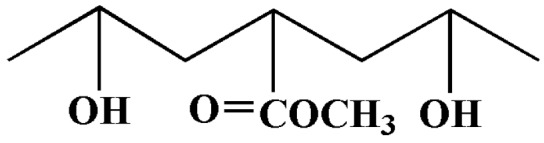
.

### 2.2. Preparation of Samples

Preparation of PVA/CMC blends: Firstly, a PVA solution was obtained by dissolving 1 g of PVA powder in 19 g of water under stirring at 90 °C for 3 h. Then, 0.4 g of CMC powder was dissolved in 19.6 g of water under stirring at 60 °C for 3 h. Finally, a homogeneous solution of PVA/CMC (2.5/1 *w*/*w*) was obtained by adding the CMC solution to the PVA solution and stirring at 85 °C for 3 h.

Preparation of PVA/CMC/Na_2_CO_3_ blends: Firstly, a PVA/Na_2_CO_3_ blend was obtained by adding an Na_2_CO_3_ solution to a PVA solution under stirring at 85 °C for 1 h. The mass of Na_2_CO_3_ was varied as 8 wt.%, 12 wt.%, and 16 wt.% of PVA. Then, a PVA/CMC/Na_2_CO_3_ solution was obtained by adding the CMC solution to the PVA/Na_2_CO_3_ solution under stirring at 85 °C for 3 h.

Preparation of composite films: Firstly, the obtained PVA, PVA/CMC, PVA/Na_2_CO_3_, and PVA/CMC/Na_2_CO_3_ film-forming solutions were ultrasonicated for 30 min. Next, films were prepared from these solutions on petri dishes using a solution-casting technique. Finally, the cast films were dried at 60 °C for 24 h. The formulations and abbreviations of the samples are shown in Table 1.

### 2.3. Fourier-Transform Infrared (FTIR) Spectroscopy

FTIR spectra were measured using a NICOLET 6700 spectrometer (Thermo Scientific Co., Waltham, MA, USA) from 4000 cm^−1^ to 400 cm^−1^ to study the interaction between PVA and CMC. Samples were ground into powder after drying.

### 2.4. X-ray Diffraction (XRD)

XRD analysis was carried out using a PANalytical X’Pert-Pro MPD (Almelo, The Netherlands) diffractometer with Cu-Kα radiation (λ = 1.5406 Å) to characterize the crystal form and crystallinity of samples. The diffraction angle ranged from 5° to 80°. Meanwhile, XRD deconvolution analysis was performed using Origin 9.0 software in order to deconvolute specific regions (crystals or amorphous peaks).

### 2.5. Differential Scanning Calorimetry (DSC)

Thermal properties were measured by DSC using a Netzsch PC200 instrument (Netzsch Inc., Selb, Germany). Specimens weighing 6–9 mg were heated at a rate of 10 °C/min from 25 °C to 250 °C under N_2_ gas purging.

### 2.6. Atomic Force Microscopy (AFM) Studies

The morphology of composite films was observed using an atomic force microscope (Veeco, DI) operated in contact mode in air. The test specimen had dimensions of approximately 1.0 mm × 5.0 mm × 5.0 mm (thickness × width × length).

### 2.7. Dielectric Properties

The dielectric properties were analyzed using a broad-peak dielectric spectrometer (Concept 40, Novocontrol, Montabaur, Germany) in the frequency range from 10^−1^ to 10^7^ Hz at room temperature. Specimens in the form of a disc of size 30 mm × 2 mm (diameter × thickness) were cut from the dried films.

### 2.8. Water Sorption

Firstly, the films were weighed (*W*_1_) and placed in 90% humidity at room temperature. Then, the films were periodically removed and weighed (*W*_2_). The experiment was repeated twice, and the average weight was determined. The water sorption was calculated as follows [19]:*Water sorption* = [(*W*_2_ − *W*_1_) × 100%]/*W*_1_.(1)

### 2.9. Solubility

Film solubility was determined using the following method: pieces of each film were firstly weighed (*W*_1_) and then immersed in deionized water at room temperature for 24 h. Next, the films were taken out and dried in an oven at 60 °C to constant weight (*W*_2_). The experiment was repeated twice, and the average weight was determined. The solubility was calculated using Equation (2) [20].
*Solubility* = [(*W*_1_ − *W*_2_) × 100%]/*W*_1_.(2)

## 3. Results and Discussion

### 3.1. FTIR Spectroscopy

Figure 1 shows the FTIR spectra of PVA, CMC, P/C, P/N, P/C/N, and Na_2_CO_3_. For pure PVA, the peaks at 3588 cm^−1^ and 1330 cm^−1^ were assigned to OH stretching and bending vibrations for PVA film [21,22]. The peak corresponding to methylene group (CH_2_) asymmetric stretching vibration appeared at −2932 cm^−1^ [23]. The peak at −1727 cm^−1^ corresponded to the C=O symmetrical stretching vibration of the unhydrolyzed ester functional group present on the PVA backbone, and the peak at −1249 cm^−1^ could be attributed to the C–O–C asymmetric stretching vibration of the ester group [24,25]. For CMC, a strong broad peak was observed at 3541 cm^−1^, which arose from OH stretching vibration [26]. The vibrational peak observed at 2915 cm^−1^ could be ascribed to CH asymmetric stretching [24]. The peak at 1607 cm^−1^ could be attributed to the asymmetric modes of stretching vibration of ester groups (COO^−^) [26]. Two absorption peaks that appeared in the region of 1417 cm^−1^ and 1325 cm^−1^ corresponded to scissoring CH_2_ and bending OH, respectively [27]. For Na_2_CO_3_, the peaks at 1426 cm^−1^, 878 cm^−1^, and 700 cm^−1^ were carbonate bands [28]. For P/C/N and P/N composite films, the absorption peaks at 878 cm^−1^ and 700 cm^−1^ indicated the presence of Na_2_CO_3_. The Na_2_CO_3_ absorption peak at 1426 cm^−1^ was not obvious, because it overlapped with the PVA absorption peak.

The FTIR spectra were analyzed based on two potential regions in Figure 2 in the following ranges of wavenumbers: (a) 3750–2750 cm^−1^ and (b) 2000–1000 cm^−1^. The spectrum of the P/C composite film exhibited, in general, the same peaks as those observed in the PVA and CMC films, indicating that no reaction occurred between PVA and CMC [29]. For P/C/N composite films, the absorption peak of OH of PVA moved further toward lower energies than those of the P/C film, exhibiting a shift from 3588 cm^−1^ to 3464 cm^−1^. The COO^−^ peak of CMC had a similar trend, shifting from 1607 cm^−1^ to1578 cm^−1^. At the same time, Figure 2b shows that the OH bending vibration peak of PVA and CMC was gradually weakened with the addition of Na_2_CO_3_. The changes in OH stretching vibration peak, OH bending vibration peak, and COO^−^ asymmetric stretching vibration indicated an enhancement of hydrogen bonding between PVA and CMC after the addition of Na_2_CO_3_, and the intensity of the absorption peak of C=O (1727 cm^−1^) of PVA gradually decreased with the increase in Na_2_CO_3_ content, which could be attributed to the hydrolysis of vinyl acetate group of PVA in an alkaline environment. The acetate group was replaced with a hydroxyl group and, thus, the absorption of C=O of the acetate group in PVA was reduced, as shown in Equation (3). The intensity of the absorption peak of the methylene group (CH_2_) for the P/C/N3 composite film was significantly reduced. Because OH is an electron-donating group, the electron cloud density around the C atom connecting the hydroxyl group was increased, and the electron cloud density around the C atom adjacent to the methylene group was affected, thus decreasing the absorption peak intensity of methylene group. For the P/N3 composite film, the absorption peak of OH moved to 3475 cm^−1^, which was related to the formation of intermolecular hydrogen bonds in PVA. The disappearance of the C=O absorption peak in the spectrum of the P/N3 composite film confirmed the reaction shown in Equation (3).

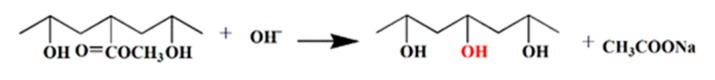
(3)

### 3.2. XRD Analysis

The XRD patterns of PVA, CMC, P/C, P/N, P/C/N, and Na_2_CO_3_ are shown in Figure 3. The crystallinity (*X_c_*) of the obtained films was determined from the equation *X_c_* = *A_c_*/(*A_c_* + *A_a_*) × 100 (where *A_c_* and *A_a_* are the peak intensities of crystalline and amorphous cellulose, respectively) [30]. The results are shown in Table 2. Figure 3 shows that pure PVA had two sharp diffraction peaks at 19.7° and 40.8° corresponding to an orthorhombic lattice (110) reflection [31]. On the other hand, three diffraction peaks for pure CMC appeared at 20.7°, 31.8°, and 45.7°. Compared to pure PVA and CMC films, the position of the diffraction peak of the P/C composite film had no obvious movement. This indicated that there was no new crystal form, but the crystallinity of the P/C composite film compared to that of the PVA film was reduced, probably because the CMC macromolecular chains inhibited the movement of the PVA molecular chains [32]. For the P/N3 film, the crystallinity was significantly weakened compared to PVA, indicating that the addition of Na_2_CO_3_ destroyed the crystal structure of PVA. After adding CMC, the crystallinity of the P/C/N composite films continued to decrease. This was related to the hindrance of the CMC macromolecular chain. However, Figure 3 and Table 2 show no significant change in the peak position and crystallinity of P/C/N composite films compared to those of the P/C film. Considering that this was related to the use of Na_2_CO_3_ as a nucleating agent, the crystallinity of P/C/N composite films was improved. Figure 3 also shows that the XRD peak corresponding to crystalline Na_2_CO_3_ was not obvious in composite films; however, it can be speculated that the crystallinity of Na_2_CO_3_ in the composite films was greatly reduced [33,34].

### 3.3. DSC Analysis

Figure 4 shows the characteristic features of the DSC curves for the studied composite films. The crystallinity of composite films could be calculated using the formula *X_c_* = (Δ*H*/*w*Δ*H_c_)* × 100%), where Δ*H_c_*= 161 J/g, and *w* is the mass fraction of PVA in the composite films [35]. The results are shown in Table 3. The crystallinity obtained from DSC was different from that obtained from XRD, but the trend of crystallinity changes was completely consistent. For the PVA film, DSC showed an endotherm melting transition at 192 °C, representing the melting temperature of PVA. The DSC thermogram of P/C showed an endotherm melting transition at 178 °C, lower than that of PVA. The melting temperature decreased due to a decrease in the crystallinity degree, consistent with the XRD and DSC results.

The P/C/N1 composite film exhibited an endotherm melting transition at 194 °C, which was higher than the melting temperature of PVA, and the crystallinity degree of the film was lower than that of PVA. This was because of an increase in the intermolecular hydrogen bonds of PVA and CMC, as indicated by the FTIR results. The hydrogen bonds between PVA and CMC inhibited the movement of PVA molecular chains, increasing the melting temperature and decreasing the crystallinity degree of P/C/N composite films. This also showed that the melting temperature of the P/C/N composite films increased with the increasing content of Na_2_CO_3_. This was because the crystal structure of PVA was gradually destroyed as the content of Na_2_CO_3_ increased. The PVA molecular chains became disordered, which strengthened the contact of PVA and CMC. At the same time, as Na_2_CO_3_ content increased, more ester groups were hydrolyzed, and the amount of hydroxyl groups increased. This generated more intermolecular hydrogen bonds and, thus, increased the melting temperature. Moreover, the crystallinity of the P/C/N composite films was higher than that of P/C composite film, consistent with the XRD results. At the same time, the crystallinity of the P/C/N composite films gradually increased with the increase in sodium carbonate content, which was also related to the role of sodium carbonate as a nucleating agent.

### 3.4. Dielectric Analysis

Figure 5 shows the patterns of dielectric constant as a function of frequency at room temperature for the resulting composite films. In general, the dielectric constant of pure PVA was low, and the dielectric constant of the P/C composite film was slightly increased. However, the dielectric constant of P/C/N composite films was significantly improved with the addition of Na_2_CO_3_, and it also gradually increased with increasing Na_2_CO_3_ content. Specially, the dielectric constant of the P/C/N3 composite film was 32,000 at 10^2^ Hz, 25 times higher than that of the P/C composite film (the dielectric constant of the P/C composite film was 1210 at 10^2^ Hz). This could be ascribed to the addition of Na_2_CO_3_, which promoted the generation of hydrogen bonds between PVA and CMC molecules. This led to a multimolecular dipole, significantly increasing the dielectric constant of P/C/N composite films.

At the same time, the dielectric constant of pure PVA slightly varied with frequency and remained constant. The change in dielectric constant for the P/C composite film was also very small. However, the dielectric constants of P/C/N composite films rapidly decreased with increasing frequency in the low-frequency range, while they decreased very slowly when the frequency was above 10^5^ Hz. This was because, with the addition of Na_2_CO_3_, the composite films had a certain dielectric relaxation phenomenon in the low-frequency range, and this phenomenon was gradually attenuated in the high-frequency range [36].

### 3.5. Water Sorption Analysis

Figure 6 shows the water sorption of PVA, P/C, and P/C/N composite films, which were significantly different. The pure PVA film exhibited the lowest water sorption. For the P/C composite film, the water sorption ability was higher than that of pure PVA. This was because the actual mixing of two different types of molecules led to the deformation of the structure, making the network more hydrophilic [29]. In addition, the poor water sorption ability of the pure PVA film could be attributed to higher intermolecular attraction and strong intramolecular or intermolecular hydrogen bonding.

The water sorption of P/C/N composite films was significantly higher than that of the pure PVA film, but lower than that of the P/C composite film. This was because the addition of Na_2_CO_3_ disrupted the crystal structure of PVA. The molecular chain of PVA changed from regular to disordered. This enhanced the contact with water molecules, making the water sorption of P/C/N composite films significantly higher than that of PVA film. Similar results were reported in the literature [18]. However, it also promoted the formation of hydrogen bonds between PVA and CMC, and the hydrogen bonds acted as a physical crosslinking point, promoting the formation of a much denser molecular structure. Therefore, its water sorption was significantly lower than that of the P/C composite film. The water sorption of the P/C/N2 composite film was lower than that of the P/C/N1 composite film because the hydrogen bonding between PVA and CMC was strengthened with the increase in Na_2_CO_3_ content, and a denser molecular structure was formed. The water sorption of the P/C/N3 composite film was slightly increased. This was related to excess Na_2_CO_3_, i.e., the PVA molecular chain was in sufficient contact with water molecules. At the same time, excess sodium carbonate also caused the ionization of the carboxyl group of CMC, and the COO/COOH ratio of CMC also increased because of the increasing ionization of carboxylic groups. This resulted in a greater mutual repulsion among the COO^−^-bearing CMC chains, consequently causing the CMC chains to undergo faster relaxation. This widened the mesh sizes of free volumes, resulting in a larger water sorption ratio [37].

### 3.6. Solubility Analysis

Figure 7 shows the solubility of PVA, P/C, P/C/N1, P/C/N2, and P/C/N3 composite films. Compared to the PVA film and P/C composite film, the solubility of the P/C/N composite films was significantly reduced with the addition of Na_2_CO_3_, further confirming the formation of intermolecular hydrogen bonding. The P/C/N2 composite film showed the lowest solubility of 24%, slightly lower than the solubility of the P/C/N1 composite film. This indicated that the hydrogen bonding network of the P/C/N2 composite film was denser. This result is similar to that of Ibrahima et al., i.e., the composite film showed the lowest solubility in a certain alkaline environment [17]. The solubility of the P/C/N3 composite film was slightly increased to 25%, which was related to the increased contact between the disordered molecular chains and water molecules. Secondly, the ionization of the CMC carboxyl group increased the mesh size of its free volume, which was also an important factor. A significant decrease in the solubility of the P/C/N composite films indicated that hydrogen bonding was the most important factor affecting the solubility of composite films, and slight changes in the solubility of the P/C/N3 composite film showed that the change in molecular structure also had a certain effect on the solubility of composite films.

### 3.7. Surface Topography Analysis

The representative images obtained from AFM studies for the studied composite films are shown in Figure 8. Clearly, film surfaces with varying levels of surface roughness were present. The surfaces of the PVA film and P/C composite film were smooth and flat, and the surface of the P/C/N1 composite film was also relatively smooth. As the content of Na_2_CO_3_ increased, the surfaces of the P/C/N2 and P/C/N3 composite films became rough and uneven; Table 4 shows the corresponding roughness. According to the results of water sorption, although the roughness of P/C/N2 increased significantly, its water sorption still decreased, indicating that the roughness of the surface of composite films had a small effect on water sorption. As the main factors of water sorption, surface roughness, pores, and molecular structure are always mentioned, obviously, no pores were found in the composite films [38]. Thus, the interaction between PVA and CMC under the action of Na_2_CO_3_ was proven to be the main factor affecting the water sorption of composite films.

### 3.8. Mechanism Analysis

FTIR, DSC, XRD, and the dielectric constant test results of P/C/N composite films were significantly different from those of the PVA film and P/C composite film. In addition, the solubility of the P/C/N composite films was much lower than that of the PVA film and P/C composite film, indicating the formation of a crosslinking structure in P/C/N composite films. The water sorption of the P/C/N composite films was higher than that of the PVA film, but lower than that of the P/C composite film, indicating that Na_2_CO_3_ promoted the relaxation of the PVA and CMC structure. Based on the above information, a reaction mechanism is proposed for Na_2_CO_3_, PVA, and CMC.

Na_2_CO_3_ is alkaline in water and ionizes OH groups. At a certain temperature (80 °C), the alkali can hydrolyze the vinyl acetate group in the partially alcoholic PVA; the acetate group is replaced with a hydroxyl group. The amount of hydroxyl groups in PVA increases, consistent with the weakening C=O absorption peak of P/C/N and P/N composite films in the FTIR spectrum. At the same time, the crystal structure of PVA is significantly destroyed in an alkaline environment, consistent with the XRD results of P/N film. The destruction of the crystal structure of PVA further stretches the molecular chain of PVA and strengthens the contact with the molecular chain of CMC, thus promoting the formation of intermolecular hydrogen bonds between the OH and COOH groups of CMC and the OH group of PVA. The shift in C=O and OH characteristic absorption peaks, the decrease in crystallinity, the increase in the melting point, and the increase in dielectric constant indicate the formation of intermolecular hydrogen bonds between PVA and CMC, as shown in Figure 9. In addition, the stretching of the PVA molecular chain enhances its contact with water molecules; the ionization of the CMC carboxyl group leads to an increase in the mesh size of its free volume, significantly increasing the water sorption of P/C composite films doped with Na_2_CO_3_. However, the increase in hydrogen bonding between PVA and CMC makes the molecular structure denser, which is the main reason for the decrease in water sorption and solubility.

## 4. Conclusions

This study shows that Na_2_CO_3_ significantly affected the microstructure and macroscopic properties of P/C/N composite films. FTIR, XRD, AFM, DSC, and dielectric properties confirmed that Na_2_CO_3_ promoted the formation of intermolecular hydrogen bonds between PVA and CMC. The water sorption and solubility tests showed that the composite films had reasonable water sorption capacity and water-resisting properties, resulting from the synergistic interaction between the relaxation of the molecular chain and intermolecular hydrogen bonding between PVA and CMC. This study showed how the salt affects the microstructure and macroscopic properties of composite films. This has long-term significance in promoting the application of biodegradable superabsorbent resins and polymer electrolytes.

## Figures and Tables

**Figure 1 polymers-12-00453-f001:**
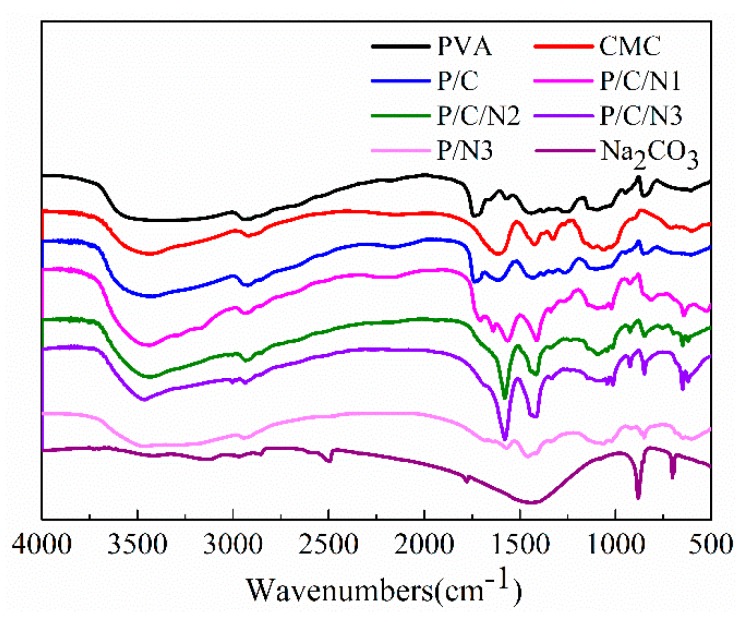
Fourier-transform infrared (FTIR) spectra of Na_2_CO_3_ and the obtained composite films.

**Figure 2 polymers-12-00453-f002:**
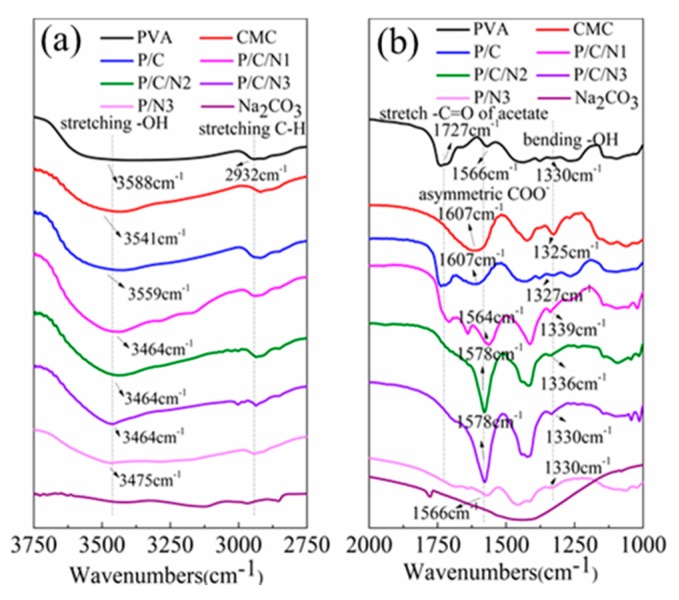
FTIR spectra of Na_2_CO_3_ and the obtained composite films: (**a**) region from 3750 cm^−1^ to 2750 cm^−1^; (**b**) region from 2000 cm^−1^ to 1000 cm^−1.^

**Figure 3 polymers-12-00453-f003:**
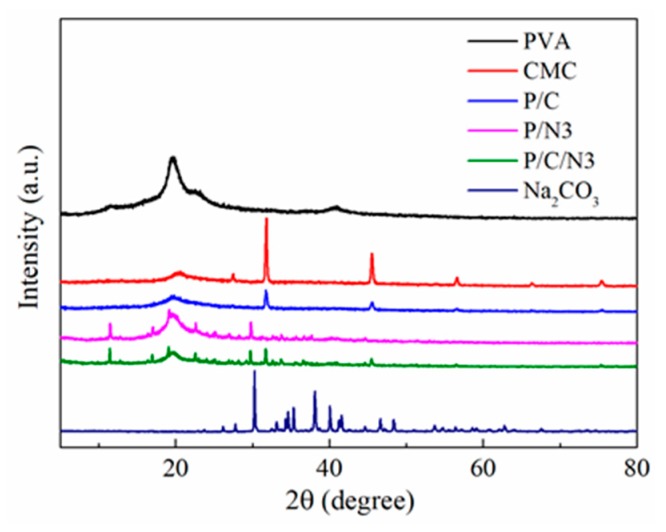
X-ray diffraction (XRD) patterns of Na_2_CO_3_ and the obtained composite films.

**Figure 4 polymers-12-00453-f004:**
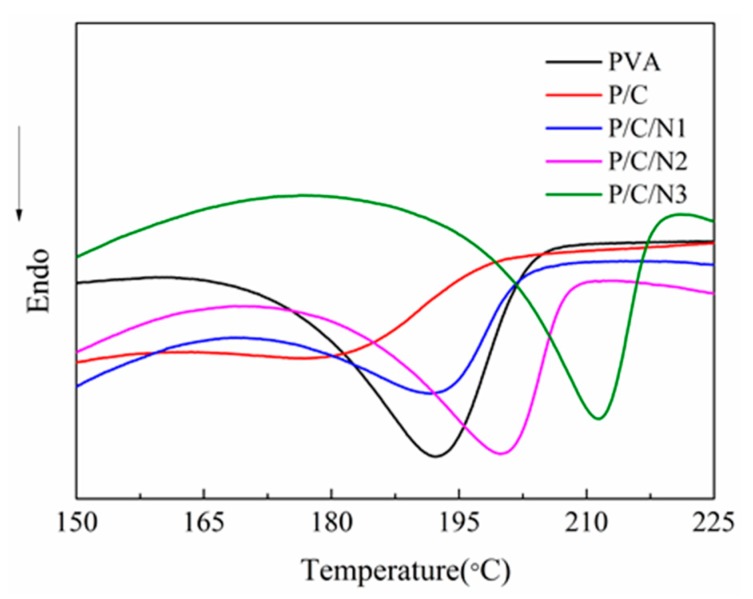
Differential scanning calorimetry (DSC) curves of the obtained composite films.

**Figure 5 polymers-12-00453-f005:**
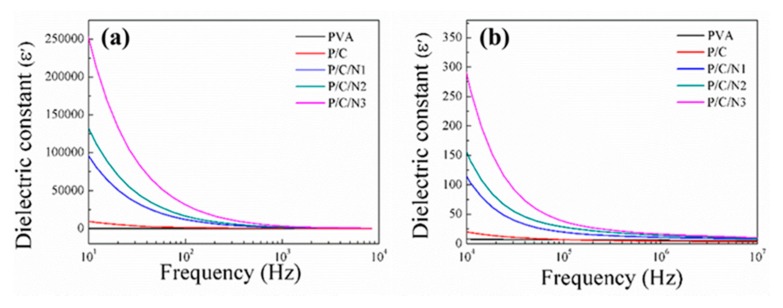
The dielectric constants of the obtained composite films at (**a**) low frequency and (**b**) high frequency.

**Figure 6 polymers-12-00453-f006:**
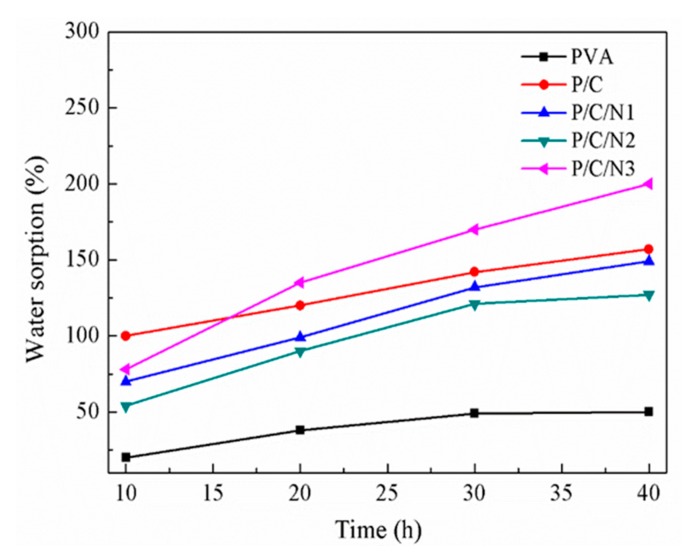
Water sorption of PVA, P/C, P/C/N1, P/C/N2, and P/C/N3 composite films.

**Figure 7 polymers-12-00453-f007:**
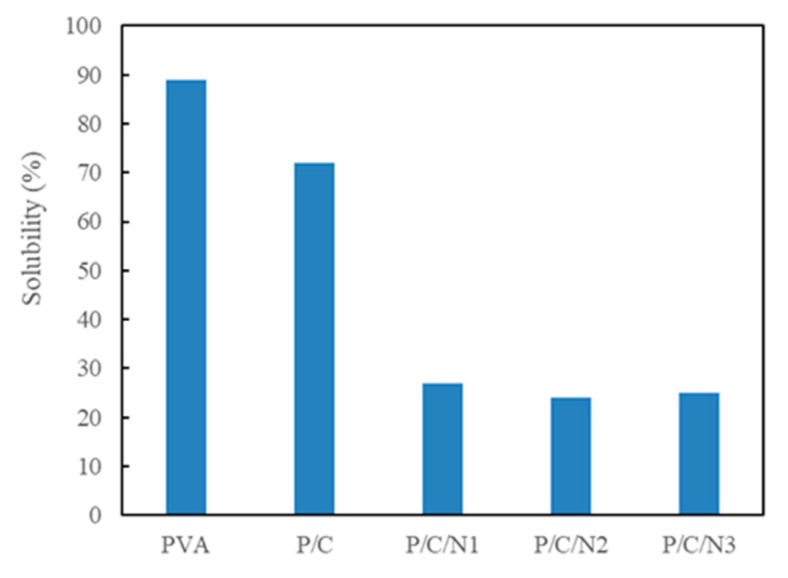
Solubility of the obtained composite films.

**Figure 8 polymers-12-00453-f008:**
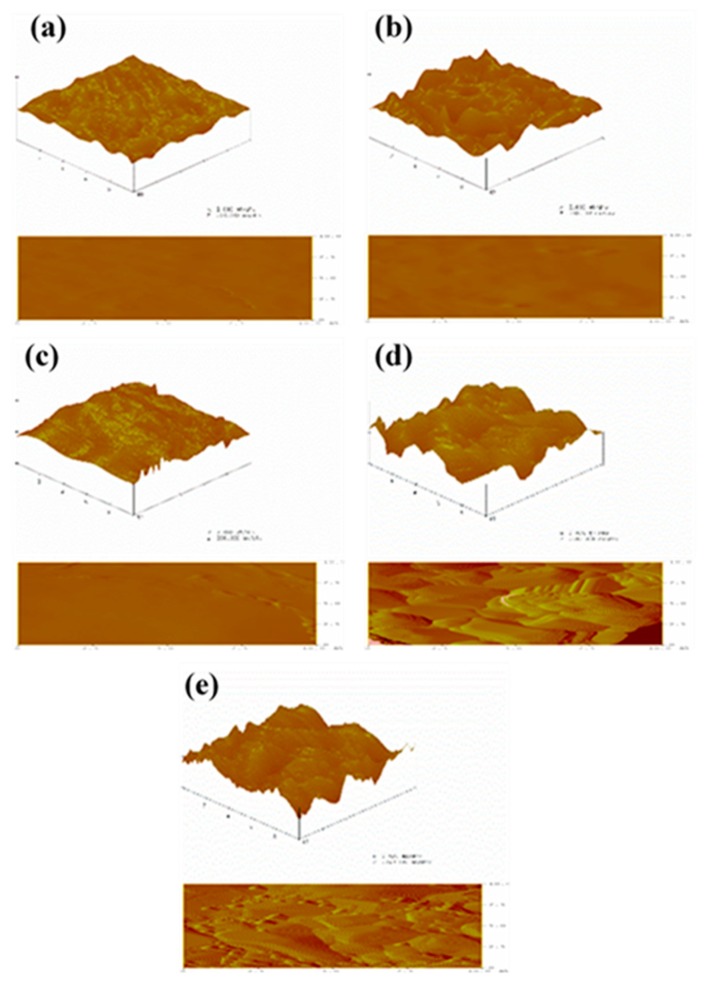
Representative atomic force microcopy (AFM) images of (**a**) PVA, (**b**) P/C, (**c**) P/C/N1, (**d**) P/C/N2, and (**e**) P/C/N3 composite films.

**Figure 9 polymers-12-00453-f009:**
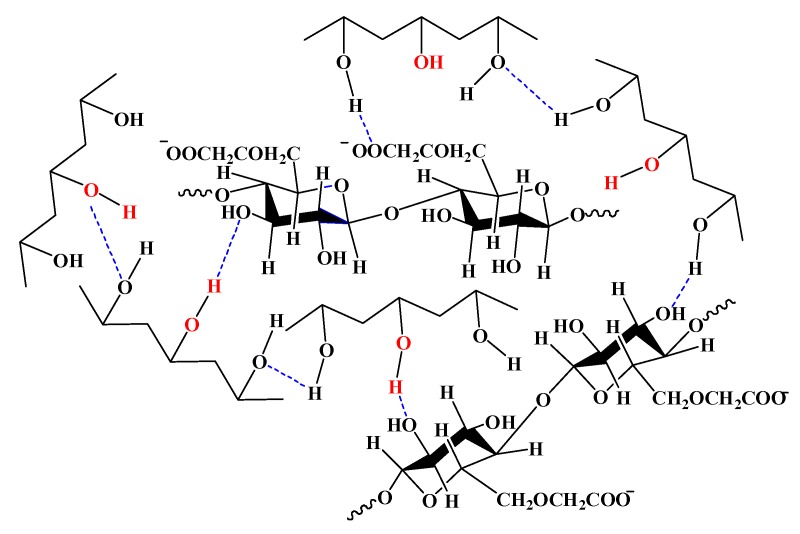
The reaction mechanism for PVA, CMC, and Na_2_CO_3._

**Table 1 polymers-12-00453-t001:** The formulations and abbreviations of the samples. PVA—polyvinyl alcohol; CMC—carboxyl methyl cellulose sodium.

Sample	Sample Notation	PVA Solution		CMC Solution		Na_2_CO_3_ Solution	
		PVA (g)	Water (g)	CMC (g)	Water (g)	Na_2_CO_3_ (g)	Water (g)
PVA	PVA	1	19				
CMC	CMC			0.4	19.6		
PVA/CMC	P/C	1	19	0.4	19.6		
PVA/CMC/Na_2_CO_3_	P/C/N1	1	19	0.4	19.6	0.08	5
	P/C/N2	1	19	0.4	19.6	0.12	5
	P/C/N3	1	19	0.4	19.6	0.16	5
PVA/Na_2_CO_3_	P/N3	1	19			0.16	5

**Table 2 polymers-12-00453-t002:** Percentage of crystallinity of the obtained composite films.

Sample	*A* *_c_*	*A* *_a_*	*X**_c_* (%)
PVA	2270	4223	34.96
CMC	373	1834	17.65
P/C	124	1536	7.47
P/C/N3	214	953	18.34
P/N3	501	1979	20.20

**Table 3 polymers-12-00453-t003:** DSC data for the obtained composite films.

Sample	PVA	P/C	P/C/N1	P/C/N2	P/C/N3
Tm (°C)	192	178	194	200	212
Melting enthalpy (J/g)	33.83	8.94	16.18	24.87	25.50
Crystallinity (%)	21.01	7.77	14.87	23.48	24.71

**Table 4 polymers-12-00453-t004:** Roughness of the obtained composite films.

	PVA	P/C	P/C/N1	P/C/N2	P/C/N3
R_q_	10.205	19.963	16.160	295.52	264.09
R_a_	7.756	15.115	11.991	233.14	210.62

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
