# Peer review of "Effect of Na2CO3 on the Microstructure and Macroscopic Properties and Mechanism Analysis of PVA/CMC Composite Film"

_polymers, 2020, doi:10.3390/polym12020453_

Round 1

Reviewer 1 Report

There are several suggestions/comments in regards to this paper :

Line 69&70- sentences need to be rechecked Table 1 - unit for water is ml or g? Line 97- please specify the humidity used in this study Line 102 - please specify the time/duration applied while immersing the sample in DI water Figure 1 - it is suggest to add the spectra for the salt (sodium carbonate) for comparisons Figure 3 - it is suggest to add the XRD profile for sodium carbonate for comparisons Figure 4 - please label (a-e) for all the XRD images Line 3.6 - Section 3.6 - the explanation for the solubility analysis is not  adequate. Is this solubility analysis also shows the shrinkage/drying ability of the films? Table 3- why P/C/N2 shows highest surface roughness?

Reviewer 2 Report

The subject could be of interest to the general readership of the journal and the authors presented a great deal of data using different characterization techniques. Here, I am summarizing my comments for a major revision consideration to improve the manuscript further:

- What type of water used to prepare the films? Please include this information in section 2.1 or 2.2.

- Please provide following information in section 2.3: how was the sample prepared, if sample was dried before the test, and no. of scans.

- Please provide following information in section 2.4: diffraction angle range, no. of scan.

- Section 3.1: Please provide appropriate references when discussing peak assignments.

- Section 3.1: Here authors have described some of their results based on peak intensity, however, it is not clear how did they normalize the spectra or by any other means to make such a comparison. Please explain.

- Section 3.1: Please include P/N3 spectrum and discuss accordingly.

- Fig 1 and 2: Please exchange the position of last to spectra (P/C/N2 and P/C/N3) to make them consistent with the legends.

- Section 3.2: Fig. 4 can be moved to supplementary information, however, please mention the software used for the peak fittings in the text.

-  Table 2: Please use similar notation for PVA/CMC

- Section 3.3: Thermogram of P/C/N3 is very confusing. Why this curve doesn’t show and peak at all? It has more % crystallinity than that of P/C as mentioned in table 2 but there is hardly any Tm. On the other hand, there is no transition temperature point either as explained by the authors for the other P/C/N samples. Please discuss.

- Fig. 5: Overall, the quality of the thermograms is very poor as the details can be hardly seen form the peaks. DSC of PVA is very common in literature with better intensity. Please replace these curves.

- Section 3.3: Calculate the % crystallinity as can be obtained from a DSC curve and compare them with the results obtained in Table 2.

- For any practical application, tensile properties of the material is very important. Please provide and discuss basic tensile data of the films compared in this manuscript.

Reviewer 3 Report

Introduction is very short, yes, a lot of references, but should be enhanced.

English can be improved.

Authors should refer their results more to the existing data. 

Round 2

Reviewer 1 Report

All given comments have been addressed accordingly by the authors in a revised manuscript.

Author Response

Thank you very much for your thoughtful advices, which have helped improve this paper substantially.